# Synthesis of Nano-Selenium from *Bombyx batryticatus* Polypeptide and Exploring Its Antioxidant and Skin Whitening Ability

**DOI:** 10.3390/molecules30051153

**Published:** 2025-03-04

**Authors:** Yang Ning, Chen Peng, Li Weihong, Feng Cuiping, Wang Xiaowen, Wang Qiling

**Affiliations:** Food Science and Engineering, Shanxi Agricultural University, Jinzhong 030810, China; chesterzuishuai@foxmail.com (C.P.); lwhsxau@163.com (L.W.); ndfcp@163.com (F.C.); wwxw11@163.com (W.X.); g19831082039@163.com (W.Q.)

**Keywords:** *Bombyx batryticatus* polypeptide, nano-selenium, antioxidant, skin whitening ability

## Abstract

To increase the stability of selenium in nano state and further improve its antioxidant and skin whitening ability, *Bombyx batryticatus* polypeptide (BBPP) was prepared. The optimum synthesis conditions of *Bombyx batryticatus* polypeptide nano-selenium (BBPP-SeNPs) were determined by a double-peak method. BBPP-SeNPs were characterized by transmission electron microscope (TEM), Fourier transform infrared spectroscopy (FT-IR), and particle size analysis (PSS). The 1,1-Diphenyl-2-picrylhydrazyl (DPPH), 2,2’-azino-bis (3-ethylbenzothiazoline-6-sulfonic acid (ABTS), superoxide anion free radical scavenging rate, and total antioxidant capacity of BBPP, vitamin C (VC), and BBPP-SeNPs were measured for comparison. The inhibitory ability of BBPP and BBPP-SeNPs on tyrosinase was measured. Using mouse modeling, the skin whitening ability of VC and BBPP-SeNPs was measured. The results showed that the optimal conditions were obtained when the concentration of BBPP was 0.16 mg/mL, sodium selenite was 0.01 mol/L, ultrasound was carried out for 30 min, ascorbic acid was added in 0.04 mol/L, and stirring temperature was 20 °C for 4 h. The antioxidant capacity of BBPP-SeNPs has significantly improved. It can be observed that BBPP-SeNPs has obvious scavenging ability on skin-reactive oxygen species through a Reactive Oxygen Species (ROS) staining section. Through Hematoxylin–Eosin (H&E) staining, it can be proven that BBPP-SeNPs has a high security threshold.

## 1. Introduction

Selenium (Se) is a trace element essential for human health [1], with various biological activities, including assisting in the treatment of cardiovascular diseases [2], enhancing immunity, and supporting anti-cancer functions [3]. Since Se cannot be synthesized in the human body, it must be obtained through dietary intake. However, due to the narrow range between the safe dose and toxic dose of Se, excessive intake can lead to Se poisoning. Se shares chemical properties with sulfur and can replace sulfur in proteins, altering their structure and function, which may cause liver damage. Symptoms of selenium toxicity include nausea, vomiting, diarrhea, hair loss, and neurological damage. The safety of Se has been a major concern. In the study by Zhang Wei et al. [4], excessive Se exposure caused liver damage and weight loss in mice within 28 days. In contrast, Wang Huali et al. [5] demonstrated through mice experiments that SeNPs exhibit higher safety thresholds. However, SeNPs are prone to aggregation due to poor stability. To construct highly stable SeNPs, surface modification using templates such as polyphenols, polysaccharides, or proteins can be employed to enhance their stability and functionality [6].

In recent years, with the increasing demand for antioxidant and skin whitening effects, skin whitening has become a research hotspot. In related studies, *Bombyx batryticatus* (BB) has frequently been reported for its skin whitening properties. A previous study [7] incorporated *Bombyx batryticatus* powder into facial masks to enhance skin whitening effects, but this approach had low utilization efficiency. *Bombyx batryticatus* is a traditional Chinese medicinal material [8], with a protein content of up to 60% of its dry weight [9]. The active components in its proteins have been shown to possess skin whitening, anti-cancer, hypoglycemic, and lipid-regulating properties [10,11,12]. However, its application is limited due to its distinctive odor. Extracting peptides from *B Batryticatus* and using them as stabilizers for SeNPs can combine the advantages of both, further enhancing antioxidant and skin whitening capabilities. Several studies have reported the use of peptides to synthesize SeNPs. For example, Tang Hongyan et al. stabilized SeNPs using tuna peptide (TP) [13], while Zeng Lixia et al. used *Nannochloropsis* peptide (AIMP) to stabilize SeNPs [14]. Both approaches demonstrated excellent stability. Bioactive peptides, characterized by their small molecular weight, high bioactivity, and multiple functional groups, can effectively stabilize SeNPs and enhance their biological activity.

To investigate the stabilizing mechanism of *Bombyx batryticatus* peptide (BBPP) on SeNPs and its skin whitening effects, proteins extracted from *Bombyx batryticatus* were enzymatically hydrolyzed into peptides and used to synthesize SeNPs. The synthesis conditions were optimized, and the resulting BBPP-SeNPs were characterized using scanning electron microscopy (SEM), transmission electron microscopy (TEM), Fourier transform infrared spectroscopy (FTIR), particle size analysis (PSS), and zeta potential analysis. The antioxidant activity of BBPP-SeNPs was compared with that of BBPP alone. Additionally, a mouse model was used to simulate skin damage induced by UV irradiation, and skin tissue sections were analyzed using ROS staining to evaluate repair effects. Liver tissue sections were also examined to assess the toxicity of SeNPs. This study provides a reference for the application of BBPP-SeNPs.

## 2. Results and Discussion

### 2.1. Optimal Conditions for the Preparation of BBPP-SeNPs

#### 2.1.1. Dual-Wavelength Method

Figure 1 shows the full-wavelength scanning absorption spectra of BBPP solution, BBPP+VC solution, VC solution, and BBPP-SeNPs solution. The solution of BBPP, VC, and BBPP+VC show no absorbance at the visible wavelength after 400 nm, while the BBPP-SeNPs solution still has a relative absorbance value after 400 nm, indicating that a large number of red SeNPs particles are generated.

According to the determination of colloid solution by the dual-wavelength method, colloid particle size parameters [15].$$B=lg(A_{2}/A_{1})/lg(\lambda_{1}/\lambda_{2})$$

A_1_ and A_2_ are the absorption values at λ_1_ and λ_2_, respectively

According to the previous experiments of Hongxu Du et al. [16], A410/A490 was used as the basis for judging the size change of SeNPs particles, and according to the formula, a higher A490/A410 ratio corresponds to smaller SeNPs size and enhanced bioactivity.

#### 2.1.2. Conditions for the Synthesis of BBPP-SeNPs

From Figure 2A, it can be seen that at the ultrasound time of 20–30 mins, A410/A490 showed an increasing trend. It can be inferred that ultrasound fully mixed the BBPP and SeNPs, the contact became close, and the stability was improved. However, after 30mins, it showed a decreasing trend, and the SeNPs formed under this condition had a tendency to increase and agglomerate. The connection between SeNPs and BBPP was broken, and some SeNPs had been agglomerated. Figure 2B shows at the stirring temperature of 0–20 °C that A410/A490 showed an upward trend, which is speculated that the particles in the solution have good activity and can form good covalent bonds because of the rising temperature. When the temperature exceeds 20 °C, the stability of the particles shows a downward trend. It is speculated that the excessive temperature causes the denaturation and folding of the peptide and fails to form good covalent bonds with SeNPs particles. From Figure 2C, it can be seen that when the stirring time is 2–4 h, the particle stability shows an upward trend. It can be inferred that under magnetic stirring, the BBPP and SeNPs particles can be fully mixed to form stable covalent bonds. When the stirring time is more than 4 h, the particle stability shows a downward trend. From Figure 2D, it can be seen that when the concentration of polypeptide is 0.08–0.16 mg/mL, the particle stability shows an upward trend, and when the concentration exceeds 0.16 mg/mL, the particle stability shows a downward trend. It is speculated that the polypeptide concentration in the solution had reached the saturation state, affecting the solution viscosity and increasing the electrostatic attraction between molecules.

It can be concluded from the results that the BBPP-SeNPs synthesized under the conditions of adding polypeptide concentration of 0.16 mg/mL, ultrasonic treatment for 30 min, stirring temperature of 20 °C, and stirring time of 4 h have good stability.

### 2.2. Characterization Results of BBPP-SeNPs

#### 2.2.1. FTIR Infrared Spectral Analysis

The infrared spectrum of BBPP shows absorption peaks at 3371 cm^−1^, 2927 cm^−1^, 1652 cm^−1^, and 1541 cm^−1^, while the spectrum of BBPP-SeNPs exhibits peaks at 3886 cm^−1^, 2920 cm^−1^, 1637 cm^−1^, and 1541 cm^−1^. The strong absorption peaks at 3371 cm^−1^ and 2927 cm^−1^ correspond to the amide A band and amide B band, respectively. In the amide A band, the peak is characteristic of –OH and –NH groups, while the corresponding peak in BBPP-SeNPs shifts to a higher wavenumber at 3886 cm^−1^. The strong absorption peaks at 1637 cm^−1^ and 1541 cm^−1^ are attributed to the amide I band and amide II band, respectively. The amide I band arises from the stretching vibration of the C=O double bond, and the amide bond, with its strong covalent nature, provides excellent stability. In BBPP-SeNPs, the amide I band shows a slight shift, while the amide II band remains unchanged. These results indicate that BBPP interacts with selenium molecules through intermolecular forces without forming new chemical bonds. Interestingly, a similar phenomenon was reported by Huang Qing et al. [17] in their study on the interaction between soybean peptides and SeNPs. This further confirms that BBPP can enhance the overall stability of SeNPs through interactions involving oxygen- and nitrogen-containing groups.

#### 2.2.2. Particle Size Analysis and Z-Potential Analysis

In order to further analyze the size and distribution of BBPP-SeNPs, zeta potential analysis and particle size analysis were carried out at room temperature 25 °C. Figure 3B shows that most of the BBPP-SeNPs particles were about 180 nm, and there was no agglomeration phenomenon, which was consistent with the results of transmission electron microscopy ensuring the activity and function of SeNPs. The potential analysis result is −20.5 mv, indicating that the BBPP-SeNPs is negatively charged, with an absolute value of >20 mv, indicating that the electrostatic repulsion between particles is relatively strong, and the particles are in a relatively stable state in solution, are not easy to sink or aggregate, and can maintain a high stability. However, when the particle size of SeNPs were detected at 7 days and 15 days, it was found that the particle size of SeNPs slightly increased, which is speculated to be a slight accumulation of partial SeNPs. In the potential analysis at 7 days and 15 days, the analysis results were −18.2 mv and −15.3 mv, respectively, indicating that BBPPP-SeNPs maintain high stability even after prolonged storage at room temperature.

#### 2.2.3. SEM, TEM Analysis

Figure 3C shows SEM images of SeNPs under different fields of view. It can be observed that SeNPs synthesized without BBPP exhibit irregular elongated and elliptical shapes with large particle sizes, reaching approximately 1 μm or even larger. The particle size of the selenium nanoparticles was not effectively controlled.

Figure 3D,E display TEM images of BBPP-SeNPs. These images reveal that the BBPP-SeNPs possess well-defined edges and are a mixture of spherical and quasi-spherical morphologies. Most SeNPs are maintained within the size range of 100–200 nm. The SeNPs appear slightly uniform but do not aggregate into larger clusters, indicating that BBPP-SeNPs can be synthesized and stabilized well in the presence of BBPP. Notably, the microscopy characterization results align with previous studies by Y. Liu et al. [18].

### 2.3. In Vitro Antioxidant Activity Analysis

From Figure 4, we know that, in the experiments of ABTS, DPPH, and superoxide anion free radical scavenging ability, there was significant improvement in the detection rate of BBPP-SeNPs compared with BBPP.

The reducing capacity of BBPP-SeNPs and BBPP was evaluated at five different concentrations, as shown in Figure 4A. The free radical scavenging potential of these compounds was evaluated based on their ability to convert ferric ions (Fe^3+^) to ferrous ions (Fe^2+^). Compared with BBPP, BBPP-SeNPS has a significant improvement.

Among them, DPPH inhibition (%) was positively correlated with BBPP-SeNPs concentration. BBPP-SeNPs showed strong antioxidant activity compared with BBPP. Two factors, the size of SeNPs and the BBPP on its surface, affected the antioxidant properties of the prepared BBPP-SeNPs [19].

Pyrogallol autoxidation can release superoxide free radicals to form colored intermediates, while antioxidants can react with superoxide free radicals to slow down the autoxidation rate, so the change of absorbance within a certain time can indicate the superoxide free radical scavenging activity of the sample. As can be seen from the figure, the scavenging ability of BBPP-SeNPS and BBPP on superoxide anion free radicals is concentration-dependent; that is, the scavenging activity of BBPP-SeNPS is significantly increased with the increase of concentration, which indicates that the combination of BBPP and SeNPs can have the scavenging ability of superoxide free radicals within a certain concentration range.

The scavenging effect of BBPP-SeNPs and BBPP at different concentrations on ABTS free radicals can be known from Figure 4D. With the increase of concentration, the scavenging rate of ABTS free radicals increased. It can be obviously seen that BBPP-SeNPS has a higher scavenging effect on ABTS free radicals than BBPP. This may be because BBPP and SeNPs combine to have stronger antioxidant capacity, which greatly enhances the antioxidant performance.

These data were consistent with those reported by Chuang Zhai et al. [20] for the synthesis of SeNPs from Cypress polysaccharide and Sumairan Bi Bi et al. [21] for the synthesis of SeNPs from extracts of Cypress and *Cinnamomum cassia*.

### 2.4. Inhibition of Tyrosine by BBPP-SeNPs

As shown in Figure 5, the inhibition ability of tyrosinase significantly increased with the increase of the concentration of SeNPs in BBPP, and the clearance rate reached 84% at 0.2 mg/mL. Compared with the inhibition ability of BBPP on tyrosinase, there was an increase of 15–20%.

### 2.5. Experimental Results from the Mouse Models

#### 2.5.1. Skin Whitening Experiment

In the experiment of skin whitening ability in mice, the mice were modeled. Skin ΔL is shown in Figure 6C. After 21 days, the ΔL in the model group was significantly higher than that in the BBPP-SeNPs group and the VC group, and the ΔL in the CK group was the lowest (*p* < 0.05). (The smaller the ΔL, the smaller the brightness change, the whiter the skin). According to the absorbance change, it can be seen that the modeling was successful. The skin brightness of the model group smeared with distilled water continued to decrease, while the skin brightness of the BBPP-SeNPs group was only slightly lower than that of the VC group. In vivo skin whitening test results show that BBPP-SeNPs has a good skin whitening effect on mouse skin and can significantly improve the black pigmentation caused by ultraviolet irradiation.

According to the fluorescence diagram of ROS staining, the accumulation of ROS in the skin of mice was different. In order to further analyze the statistical data in Figure 6F analyzed by ImageJ software (https://imagej.net/ij/download.html, accessed on 22 February 2025), ROS in the BBPP-SeNPs group, VC group, and CK group decreased significantly compared with the H_2_O group, and the CK group was not subjected to ultraviolet irradiation, indicating that both BBPP-SeNPs and VC have obvious scavenging ability for ROS, and BBPP-SeNPs and VC have similar scavenging ability for ROS. The result indicated that BBPP-SeNPs had a strong scavenging ability for ROS.

#### 2.5.2. Toxicity Experiment

As shown in the Figure 7, both the CK group and the BBPP-SeNPs group exhibited a gradual increase in body weight over time, which is attributed to normal growth in mice. Notably, no significant weight loss was observed in the BBPP-SeNPs group, indicating no systemic toxicity associated with the treatment.

Comparative analysis of H&E-stained liver sections between the BBPP-SeNPs and CK groups revealed no structural abnormalities, inflammatory infiltration, or tissue damage in the treated mice. Key observations include: normal hepatocyte morphology with intact cellular membranes and nuclei; preserved portal areas without signs of inflammation or necrosis; no pathological alterations in liver architecture, confirming the absence of hepatotoxicity.

By observing H&E-stained epidermal sections from both groups, it was found that the BBPP-SeNPs group exhibited an intact epidermal layer with uniform keratinocyte arrangement and normal distribution of dermal collagen fibers. No signs of hyperplasia, atrophy, necrosis, or inflammatory cell infiltration were observed, indicating that SeNPs did not induce acute or chronic inflammatory responses in the skin. Hair follicles, sebaceous glands, and sweat glands retained their typical morphology and density, suggesting no adverse effects of SeNPs on skin appendages.

These experimental results ultimately demonstrate that, under the tested conditions, nano-selenium caused no structural or inflammatory alterations in mouse skin and liver. This highlights its potential as a safe nanomaterial for biomedical applications, particularly in scenarios requiring direct contact with skin tissues.

## 3. Discussion

Previous studies have found that BB has significant therapeutic potential in skin whitening [22]. However, due to its unique odor and low utilization efficiency, its application in skin whitening has remained limited to supplementary treatment. Recent research [13] has discovered that peptides can act as stabilizers to modify SeNPs, enhancing the bioactivity and utilization of both. Given that BBPP itself possesses excellent antioxidant properties, its combination with SeNPs can further enhance antioxidant activity. Additionally, SeNPs can achieve higher stability and bioactivity through BBPP. Therefore, this experiment investigated the antioxidant and skin whitening capabilities of BBPP-SeNPs through the combination of BBPP and SeNPs.

In our experiment, we first extracted the most abundant active substance from BB, BB protein, and obtained BBPP through enzymatic hydrolysis. BBPP was then synthesized with sodium selenite to form SeNPs. The optimal conditions for preparing SeNPs were determined using the dual-wavelength method. It was found that the most stable BBPP-SeNPs were obtained under the conditions of 4 h of stirring, a temperature of 20 °C, the addition of 0.16 mg/mL BBPP, and 30 min of ultrasonication. SEM was used to analyze SeNPs without BBPP, while TEM was used for BBPP-SeNPs. The comparison revealed that SeNPs without BBPP had excessively large particle sizes and poor stability, whereas BBPP-SeNPs exhibited particle sizes mostly between 100 and 200 nm, demonstrating high stability and dispersibility in solution. These results indicate that BBPP plays a stabilizing role for SeNPs. Further analysis included PSS and zeta potential measurements of BBPP-SeNPs, which were stored at 25 °C and tested every 7 days. The median particle diameter of BBPP-SeNPs was around 180 nm, with a zeta potential of −20.5 mV, consistent with the TEM results. The BBPP-SeNPs maintained good stability even after 7 and 15 days. Infrared spectroscopy was also conducted on SeNPs, as it is a crucial method for identifying interactions between substances. Previous studies [14] have shown that when peptides combine with SeNPs, the interaction primarily occurs through hydrogen bonds on hydroxyl groups and amino acids, as well as C=O bonds, with characteristic peaks at 3383, 1560, and 1664 cm^−1^. Interestingly, similar phenomena were observed in this experiment. By comparing the infrared spectra of BBPP and BBPP-SeNPs, shifts in the amide A band, amide B band, and amide I band were noted. BBPP contains unique amino acid compositions and abundant active groups, exhibiting strong reducibility and complexation. When mixed with selenite solution, SeO3^2−^ is reduced to selenium atoms, undergoing complex changes. In the initial stages of nano-selenium formation, the nanoparticles are small with high surface free energy, and their aggregation can significantly lower the system’s energy, allowing SeNPs aggregate to exist stably. However, as aggregation continues, the SeNPs grow larger, reducing their surface free energy. The energy released from aggregation becomes less than that released from the complexation of SeNPs with biomolecules, leading to the stabilization and improved dispersibility of SeNPs in the reaction solution during the middle and late stages of synthesis. Studies using dispersants or stabilizers to prepare SeNPs, such as those by Mu Jingjing et al. [22], have demonstrated that Se can interact with –NH_2_, –COOH, –SH, and –OH groups on protein molecules, which aligned with our findings.

In this experiment, we investigated the antioxidant activity and skin whitening ability of BBPP-SeNPs. In terms of antioxidant properties, BBPP-SeNPs showed varying degrees of improvement compared to BBPP alone. For skin whitening, tyrosinase is a key enzyme in the human body that catalyzes the production of melanin from tyrosine, making the evaluation of tyrosinase inhibition crucial. BBPP-SeNPs demonstrated a significant enhancement in tyrosinase inhibition compared to BBPP, effectively reducing melanin production in humans. In addition to in vitro experiments, we also conducted mouse modeling studies. On one hand, we shaved the backs of mice and exposed them to UV light to simulate sun-induced skin damage. The mice were then divided into groups and treated to study the skin darkening repair ability and ROS clearance ability of BBPP and BBPP-SeNPs. On one hand, to simulate sunlight-induced skin damage, mice were shaved on their backs and exposed to UV lamp irradiation. They were then divided into groups and administered BBPP or BBPP-SeNPs to evaluate their skin-darkening repair capacity and ROS scavenging ability in the skin. Experimental results demonstrated that BBPP-SeNPs exhibited superior skin whitening and ROS repair capabilities, slightly lower than VC. However, VC is prone to oxidation under light or air exposure during use [23], showcasing a lack of stability, whereas BBPP-SeNPs showed significantly better stability than VC, as illustrated in Figure 3B. On the other hand, after administering BBPP-SeNPs to mice, changes in body weight were recorded, and H&E-stained sections of skin and liver tissues were observed. The results revealed that BBPP-SeNPs had a high safety threshold, causing no structural or inflammatory alterations in the skin or liver.

## 4. Materials and Reagents

### 4.1. Extraction of Bombyx batryticatus Polypeptide

Slightly modified with reference to a previous method of ultrasonic protein extraction [24], specifically, using 45% ethanol with a solid–liquid ratio of 1 g: The temperature of 25 mL was 30 °C, the ultrasonic wave was 40 khz, and the power was 400 w. After the extracted protein was vacuum-lyophilized, the BB protein powder was obtained. With reference to the method of enzymatic hydrolysis of protein by Wang Yuxia et al. [25] and with slight modifications, the ratio of sample to buffer solution (S/L) was 1:1 *w*/*v*, the ratio of enzyme to substrate (E/S) was 3% (*w*/*w* protein), the hydrolysis temperature was 30 °C, and the extracted BB protein was freeze-dried under vacuum to obtain the BBPP powder.

### 4.2. Preparation of BBPP-SeNPs

According to the synthesis method of Jiang Wenyi et al. [26] with slight modifications, 5.0 mL of BBPP solution of different concentrations (0.08 mg/mL, 0.12 mg/mL, 0.16 mg/mL, 0.2 mg/mL, 0.24 mg/mL) was combined with 5.0 mL Sodium selenite (0.01 mol/L), which was then mixed evenly and treated with ultrasound (80 W, 40 kHz) for a period of time (20 min, 30 min, 40 min, 50 min). Then 5.0 mL newly prepared 0.04 mol/L ascorbic acid solution was slowly added and reaction time (2 h, 3 h, 4 h, 5 h, 6 h) was obtained at different temperatures (0 °C, 25 °C, 40 °C, 60 °C, 80 °C). The solution was washed with ultra-pure water, repeated three times, and absorbance scanning was carried out on the obtained solution, with the bimodal ratio being used as the basis for the change of SeNPs particle size. The dual-wavelength method [27] was used to determine the stability of the synthesized SeNPs and the optimal synthesis conditions. Each sample was repeated three times.

### 4.3. Characterization Method of BBPP-SeNPs

#### 4.3.1. Fourier Infrared Spectrum Analysis

The samples (BBPP and BBPP-SeNPs) were mixed with potassium bromide at a ratio of 1:100, ground in a mortar to 2 mm, and then transferred to a mold, pressing the tablets at 20 MPa pressure for about 0.5 min. Infrared spectral scanning was performed in the range of 4000~400 cm^−1^.

#### 4.3.2. Particle Size and Zeta Potential Analysis

BBPP-SeNPs was prepared into 0.25 mg/mL solution, and after being overflowed with 0.45 μm aqueous membrane, particle diameter and Zeta potential were determined by particle size and using a potential analyzer. The samples were then stored at room temperature and repeated tests were performed on 7 and 15 days.

#### 4.3.3. TEM and SEM Scan Analysis

BBPP-SeNPs was appropriately diluted with deionized water and ultrasounded, after which a few drops were placed on a copper grid coated with carbon film, spread out and dried at room temperature, and the morphology, size, and uniformity of BBPP-SENPs were observed by transmission electron microscopy at an accelerated voltage of 100 kV.

The newly prepared 1 mg Se without BBPP was pasted on the conductive adhesive and sprayed with gold. Then the morphology and structure of the samples were observed by scanning electron microscopy under 5 kV voltage.

### 4.4. Determination of Antioxidant Activity In Vitro

#### 4.4.1. DPPH Radical Scavenging

The DPPH radical scavenging activity was measured according to the method described in reference [28]. A DPPH-ethanol solution (2 × 10^−4^ mol/L) was prepared and transferred into a brown bottle. In a dark room, 2 mL of BBPP or BBPP-SeNPs solutions at concentrations of 20, 40, 60, 80, and 100 µg/mL were added to test tubes, followed by 2 mL of the DPPH-ethanol solution. The mixtures were shaken thoroughly and reacted in a 37 °C water bath for 30 min. The absorbance of the sample (A_s_) was measured at 517 nm. For the control group (A_0_), 2 mL of distilled water was used instead of the sample solution and mixed with 2 mL of the DPPH-ethanol solution, followed by the same reaction and measurement steps. Each concentration gradient of the sample solution was tested in triplicate. The DPPH radical scavenging rates of BBPP-SeNPs and BBPP were calculated using the following formula:$$DPPH\;\mathrm{Scavenging}\;\mathrm{rate}\left(\%\right)=\frac{\left(A_{0}-A_{s}\right)}{A_{0}}\times 100\%$$

A_0_—Absorbance value of the blank solution;

As—Absorbance value of the sample solution.

#### 4.4.2. ABTS Radical Scavenging

The ABTS radical scavenging activity was determined according to reference [29]. Briefly, the BBPP-SeNPs solution was mixed with the ABTS solution and reacted under light-protected conditions for 10 min. The absorbance was measured at 734 nm. Each concentration gradient of the sample solution was tested in triplicate, and the ABTS radical scavenging rate was calculated.$$ABTS\;radical\;scavenging\;rate\left(\%\right)=\frac{\left(A_{0}-A_{s}\right)}{A_{0}}\times 100\%$$

A_0_—Absorbance value of the blank solution;

A_s_—Absorbance value of the sample solution.

#### 4.4.3. Superoxide Anion Radical Scavenging

The superoxide anion scavenging activity was measured according to reference [30]. Briefly, 1 mL of BBPP, VC, or BBPP-SeNPs solutions at concentrations of 20, 40, 60, 80, and 100 µg/mL were mixed with 4.5 mL of Tris-HCl buffer (pH 8.2) and incubated in a 25 °C water bath for 20 min. Subsequently, 0.2 mL of pyrogallol solution (3 mmol/L) was added. After thorough mixing, the reaction was continued in the 25 °C water bath for 5 min, followed by termination with HCl solution (8.0 mmol/L). The absorbance of the sample was measured at 325 nm. Each concentration gradient of the sample solution was tested in triplicate. The superoxide anion scavenging rate was calculated using the formula:$$Superoxide\;anion\;scavenging\;rate\left(\%\right)=\frac{\left(A_{0}-A\right)}{A_{0}}\times 100\%$$

A_0_—Absorbance value of the blank solution;

A_s_—Absorbance value of the sample solution.

#### 4.4.4. Reducing Power

The method for determining the restoration power was adapted from the approach described by Qian Bai et al. with minor modifications [31]. The reducing power was determined as follows: 1 mL of BBPP or BBPP-SeNPs solutions at concentrations of 20, 40, 60, 80, and 100 µg/mL were mixed with 2.5 mL of PBS (20 mmol/L, pH 6.6) and 2.5 mL of potassium ferricyanide solution (1%, *w*/*v*). The mixture was thoroughly vortexed and incubated in a 50 °C water bath for 20 min. Subsequently, the reaction solution was immediately cooled on ice for 5 min, followed by the addition of 2.5 mL of trichloroacetic acid solution (10%, *v*/*v*). After centrifugation, 2.5 mL of the supernatant was collected and mixed with 2.5 mL of distilled water and 0.5 mL of FeCl_3_ solution (0.1%, *w*/*v*). The mixture was allowed to react at room temperature for 10 min, and the absorbance was measured at 700 nm. The absorbance was used to indicate the reducing power. Each concentration gradient of the sample solution was tested in triplicate.

### 4.5. Determination of Whitening Ability In Vitro

Human skin color is mainly determined by the content and distribution of melanin in the skin, and the production mechanism of melanin is generally believed to be the synthesis of tyrosine in the melanosomes of melanocytes through tyrosinase. In the process of skin melanin production and metabolism, melanin production is the most important step, and tyrosinase is the key enzyme of melanin production, which controls the formation of melanin. Since tyrosinase plays a key rate-limiting role in melanin biosynthesis, tyrosinase inhibitors have become the most important targets for the treatment of pigment-related diseases [32]. Since tyrosinase can catalyze L-dopa to produce dopa pigment, its product has a characteristic absorption peak at λ = 475 nm, and then the activity of tyrosinase can be measured. The inhibition rate of tyrosinase activity of the sample was calculated by adding the sample into the reagent. Therefore, the tyrosinase inhibition ability of BBPP-SeNPS and BBPP could be determined by referring to the determination of the oxidation rate of dopa pigment [33], and the method was slightly changed. As shown in the Table 1, 200U/ML tyrosinase solution and 1.01 mmol/L L-Dopa solution were prepared with PBS solution. The four groups of samples A1, A2, A3, and A4 were accurately absorbed in test tubes according to the Table 1, kept at constant temperature in 37 °C water bath for 10 min, and 0.5 mL 1 mmol/L L-dopa solution were added, respectively. Immediately after the reaction, the absorption value at 475 nm was recorded, and the tyrosinase inhibition rate was calculated according to the formula.$$Tyrosinase\;inhibition\;rate=\frac{\left(\begin{array}{c} A_{3}-A_{4} \end{array})-(\begin{array}{c} A_{1}-A_{2} \end{array}\right)}{A_{3}-A_{4}}\times 100\%$$

### 4.6. Experimental Methodology in Mouse Models

#### 4.6.1. Animal Ethics Committee Statement

All animal procedures and experimental protocols were approved by the Laboratory Animal Ethics Committee of Shanxi Agricultural University (Approval No. SYXK (Jin) 2020-0003).

#### 4.6.2. Experimental Methodology for Skin Whitening 

##### Grouping and Modeling in Skin Whitening

The skin whitening test in vivo was based on the method previously reported by Hsieh et al. [34], with some minor modifications. Male mice were purchased from the Beijing Laboratory Animal Center. After one week of adaptive feeding, 24 mice were randomly divided into 4 groups, namely the model group, the blank group, the BBPP-SeNPS group and the positive group. An area of 1.5 cm × 1.5 cm was removed from the back of the mice by felling, and the area of the blank group, BBPP-SeNPS group and positive group were irradiated by UVB light. The lamp was about 15 cm away from the irradiated area; irradiation was conducted every alternate day for three weeks.

##### Administration in Skin Whitening

After each irradiation, distilled water was applied behind the blank group, 5% vitamin C aqueous solution was applied behind the positive group, and 5% BBPP-SeNPS solution was applied behind the BBPP-SeNPs group. Each dose was 2 mL for 30 min.

##### Brightness Variation

Colorimeter measurements were performed on the posterior area of each mouse before the first irradiation and after each irradiation, and the change value ΔL was calculated.

##### Slice Observation

The skin of mice was compared by embedding section and ROS staining.

#### 4.6.3. Experimental Methodology for Toxicity 

##### Grouping and Modeling in Toxicity

Male mice were purchased from the Beijing Laboratory Animal Center and acclimated for one week under controlled conditions (50 ± 10% humidity, 23 ± 2 °C, 12-h light/dark cycle). After acclimatization, the mice were randomly divided into two groups: the observation group (BBPP-SeNPs treatment) and the CK group (distilled water treatment). A 1.5 cm × 1.5 cm area on the dorsal skin of each mouse was shaved to establish a standardized topical application site for dosing.

##### Administration in Toxicity

The observation group received topical applications of BBPP-SeNPs at a dose of 5000 mg/kg body weight, while the CK group received equivalent volumes of distilled water. After 24 h of initial application, the treated area was rinsed with distilled water. Subsequent applications were administered every other day for three weeks. If mortality occurred at the limit dose, the LD_50_ was determined using Horn’s method.

##### Weight and Biopsy

Body weight was recorded during each administration to monitor potential toxicity-related changes. After three weeks, all mice were euthanized and liver tissues were collected for histopathological examination. Tissue sections were analyzed under a microscope to evaluate hepatotoxicity, including structural abnormalities, inflammatory infiltration, or necrosis.

### 4.7. Data Processing

IBM SPSS Statistics 26 was used to conduct orthogonal design and variance analysis on the data results. Ducan’s multiple comparison method was used to compare and analyze the results of the single-factor experiment. The results were expressed as “mean ± standard error”, and *p* < 0.05 indicated significant difference, using origin 2024 for image rendering.

## 5. Conclusions

In summary, a novel and highly safe BBPP-SeNPs was successfully synthesized using BBPP as a stabilizing agent. Compared to BBPP alone, BBPP-SeNPs exhibit significantly enhanced antioxidant and skin whitening capabilities. These findings suggest that BBPP-SeNPs can serve as an excellent antioxidant and skin whitening agent, with potential applications in healthcare, food preservation, and environmental remediation.

## Figures and Tables

**Figure 1 molecules-30-01153-f001:**
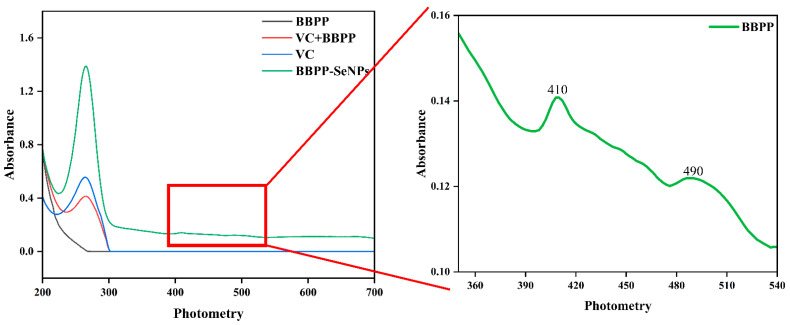
Full-wavelength of BBPP, BBPP+VC, BBPP-SeNPs.

**Figure 2 molecules-30-01153-f002:**
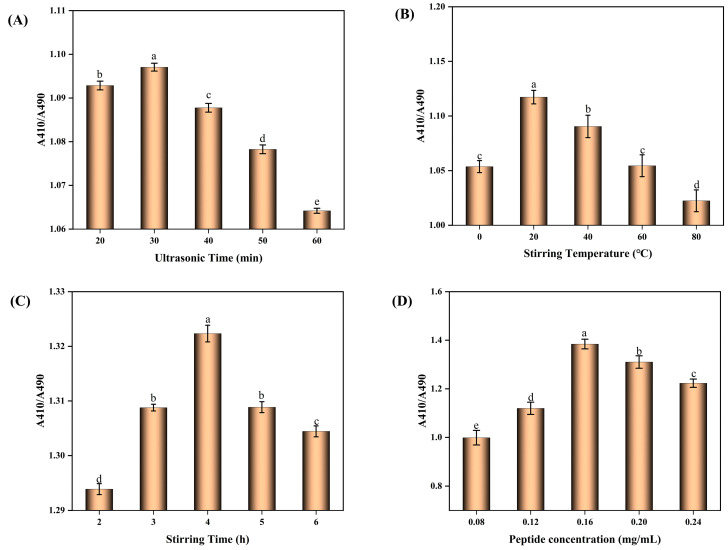
Conditions for the synthesis of BBPP-SeNPs. Each sample was tested three times. The lowercase letters “a–e” indicate a statistically significant difference, denoting superiority over other conditions. *p* < 0.05.

**Figure 3 molecules-30-01153-f003:**
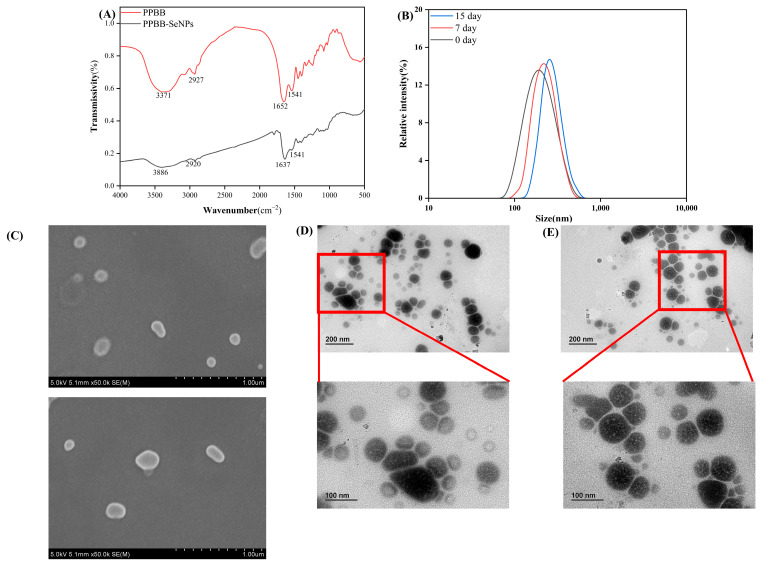
(**A**) FTIR spectra of BBPP and BBPP-SeNPs, (**B**) PSS analysis of BBPP-SeNPs in 15 days, (**C**) SEM image of SeNPs without BBPP, (**D**,**E**) TEM image of BBPP-SeNPs under different fields of view.

**Figure 4 molecules-30-01153-f004:**
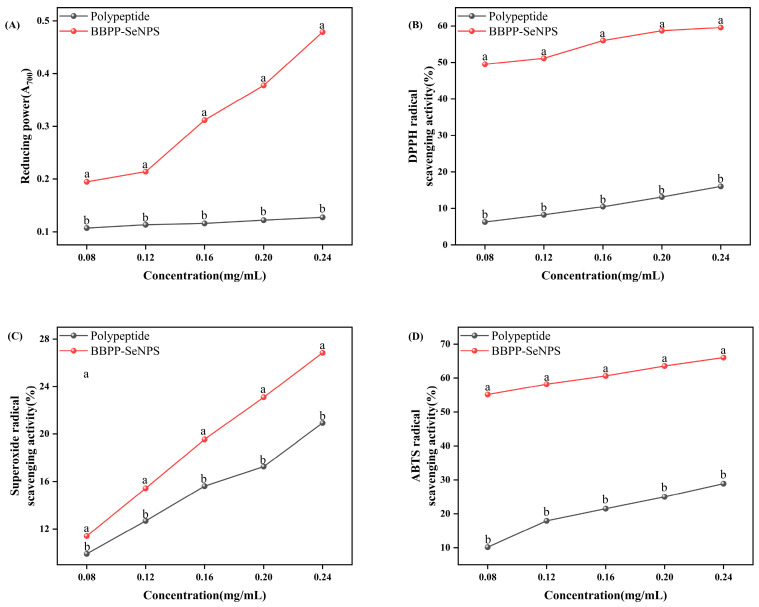
In vitro antioxidant activity analysis. Each sample was tested three times. The lowercase letters “a, b” indicate a statistically significant difference, denoting superiority over other conditions. *p* < 0.05.

**Figure 5 molecules-30-01153-f005:**
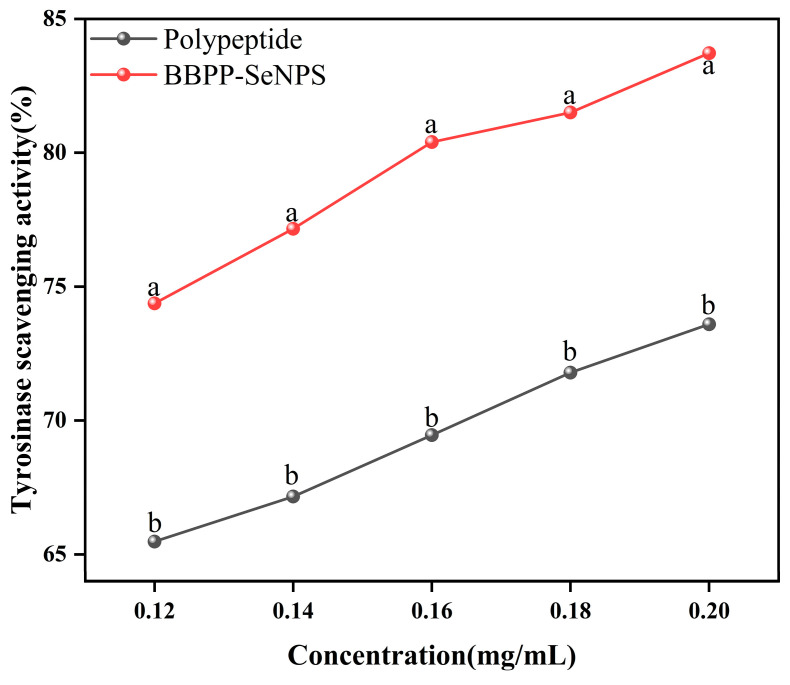
Inhibition of tyrosine by BBPP-SeNPs and BBPP. Each sample was tested three times. The lowercase letters “a, b” indicate a statistically significant difference, denoting superiority over other conditions. *p* < 0.05.

**Figure 6 molecules-30-01153-f006:**
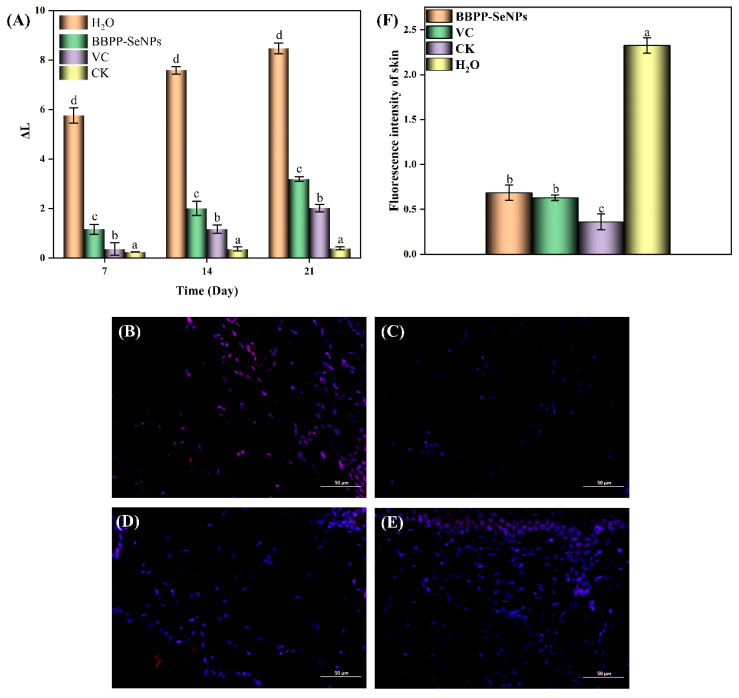
(**A**) Change of ΔL, (**B**) BBPP-SeNPs group ROS fluorescence staining image, (**C**) CK group ROS fluorescence staining image, (**D**) VC group ROS fluorescence staining image, (**E**) H_2_O group ROS fluorescence staining image, (**F**) Analysis by ImageJ software. Each sample was tested three times. The lowercase letters “a–d” indicate a statistically significant difference, denoting superiority over other conditions. *p* < 0.05.

**Figure 7 molecules-30-01153-f007:**
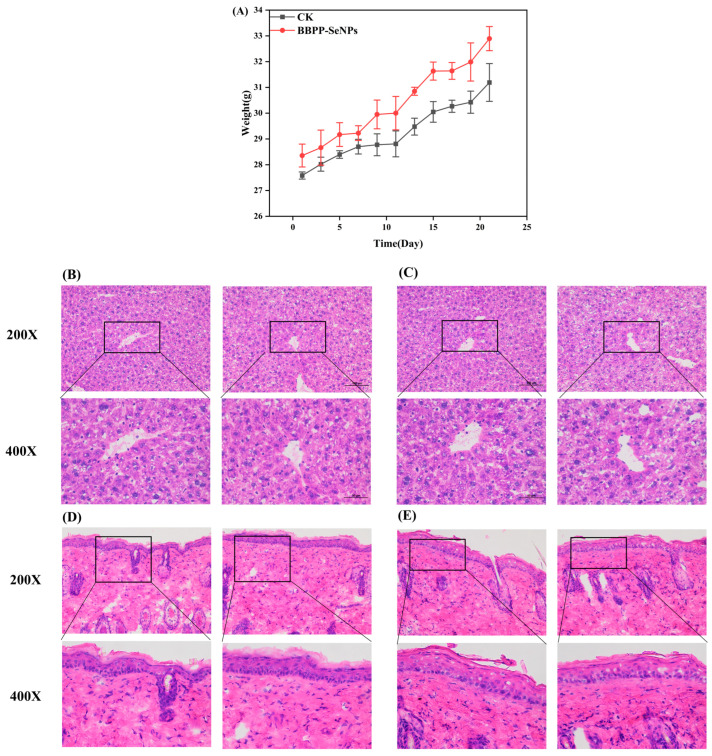
(**A**) Weight change in mice, (**B**) CK group liver, (**C**) BBPP-SeNPs group liver, (**D**) CK group skin, (**E**) BBPP-SeNPs group skin.

**Table 1 molecules-30-01153-t001:** Solution Concentration Across Experimental Groups.

Group.	V (Sample Solution)/mL	V (PBS)/mL	V (Tyrosinase)/mL
A_1_	0.5	1.5	0.5
A_2_	0.5	2	0
A_3_	0	2	0.5
A_4_	0	2.5	0

## Data Availability

The data presented in this study are available on request from the corresponding author.

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
