# Peer review of "Synthesis of Nano-Selenium from Bombyx batryticatus Polypeptide and Exploring Its Antioxidant and Skin Whitening Ability"

_molecules, 2025, doi:10.3390/molecules30051153_

Round 1
Reviewer 1 Report
Comments and Suggestions for Authors
The manuscript presents several issues that need to be addressed for clearer presentation and stronger data interpretation. Firstly, while BBPP-SeNPs are claimed to be synthesized, there is insufficient evidence to confirm nanoparticle formation. Characterization techniques such as TEM, FT-IR, are mentioned, but the results are unclear or not sufficiently discussed, and more detailed data, particularly TEM images showing particle size and distribution, as well as FT-IR spectra with interpretations of functional group interactions, are needed to confirm the nanoparticle formation. Secondly, the methodology for assessing tyrosinase inhibition is vague. Moreover, the optimal synthesis conditions for BBPP-SeNPs are determined using the "double-peak method," but this technique is not explained, leaving its relevance unclear. The manuscript should provide a brief explanation of this method and present supporting data for the chosen conditions. Finally, the lack of statistical analysis throughout the manuscript makes it difficult to assess the significance of the results, and inclusion of appropriate statistical tests is necessary to validate the conclusions. Overall, substantial revisions are needed to improve clarity, strengthen the methodology, and ensure the reliability of the findings.
Reviewer 2 Report
Comments and Suggestions for Authors
Dear Authors,
I have carefully reviewed your manuscript, titled "Synthesis of nano-selenium from Bombyx Batryticatus polypeptide and exploring its antioxidant and whitening ability " (molecules-3437350).
Although the general topic of the manuscript may fit the scope of Molecules, I believe that it may be better suited to a materials journal. In any case, the manuscript needs major changes before it can be accepted for publication. The writing, formatting, and some other errors found during the review, make me think that this version is at an early stage to be published in Molecules.
Below I list some areas of opportunity found in the text, which could improve the quality of the text:
1. In the abstract (but also in the rest of the manuscript), acronyms are used without first describing their meaning. For example, TEM, FT-IR, PSS, ABTS, etc. This should be reviewed and corrected promptly throughout the manuscript.
2. In several parts of the text (Introduction and Materials and reagents), the references are incorrectly cited. I mean that the authors mention the full name of the first author of the citation, when they should include the last name followed by et al.
3. The reference list is also poorly written. Here the authors use "et al." to mention when there are more than 3 authors in the citation. This should be corrected.
4. The introduction needs a good justification, which goes beyond mentioning the benefits of the Bombyx Batryticatus plant.
5. The term "whitening" should be described in detail, so that it is understood that it refers to skin whitening.
6. The section 2 (Materials and reagents) is poorly described. For example, the authors only mention the characterization techniques but do not provide any details on how they were performed.
7. In this same section, they make excessive use of "in short" to say "briefly." In any case, not much detail is given about the experiment.
8. In general, the results may be interesting, but they lack discussion. This is mandatory for any scientific article.
9. The authors base most of their results on the absorbance results and on the calculation of the colloidal particle size parameters (B) from the bimodal model. For this, they use the absorbance intensities at 410 and 490 nm wavelengths. However, it is not clear why these two wavelengths were selected, since the absorbance spectrum shows that the particles (VC, VC+BBPP, BBPP-SeNPs) absorb at a wavelength of 265 nm. While in the visible range there is no absorption.
10. In general, the size of most of the figures is small, which makes them difficult to read.
11. In the FTIR characterization part, the authors speculate about the formation of new bonds, which is not easy to demonstrate using this technique.
12. Regarding the TEM micrographs, the authors mention that "BBPP can protect SeNPs well." This is understood as core-shell particles; however, the TEM images do not allow us to see that the SeNPs are actually coated with organic material. As a suggestion, to corroborate this, you can include higher magnification images and an element mapping where Se, C and O are included. Obviously, avoid using continuous carbon membrane grids.
13. The authors use a mouse model to perform whitening tests. Although they provide an understandable description of this procedure, the text lacks the Institutional Review Board Statement. Please correct this.
14. The whitening results (Delta L) show (and this is mentioned by the authors) that the blank or model group (to which only water was applied after exposure to UV light) presented the highest Delta L or whitening values. In second place was the group that BBPP-SeNPs was applied. So, the effect of these particles is not as evident as using only water. This should be explained in detail.
Comments on the Quality of English LanguageThese were included in the "Comments and Suggestions for Authors
".
Reviewer 3 Report
Comments and Suggestions for Authors
This article meticulously investigates the synthesis of selenium nanoparticles (SeNPs) derived from polypeptides (BBPP) extracted from Bombyx Batryticatus, with a particular emphasis on evaluating their antioxidant and whitening properties. The study employs a rigorous double-peak method to ascertain the optimal synthesis conditions for BBPP-SeNPs. Comprehensive characterization of these nanoparticles was conducted using advanced techniques such as transmission electron microscopy (TEM), Fourier-transform infrared spectroscopy (FT-IR), and particle size analysis (PSS). Furthermore, the article presents a comparative analysis of the antioxidant capacities of BBPP, vitamin C (VC), and BBPP-SeNPs, alongside an assessment of their inhibitory effects on tyrosinase activity. The whitening efficacy of VC and BBPP-SeNPs was also examined in a mouse model.
The strengths of this research are manifold:
1. The innovative integration of traditional Chinese medicinal ingredients with modern nanotechnology offers novel insights into the development of functional materials.
2. The experimental design for determining the synthesis conditions of BBPP-SeNPs is both logical and robust, utilizing the bimodal method to pinpoint the optimal parameters. The thorough characterization via TEM, FT-IR, PSS, and other analytical techniques underscores the rigor of the study.
3. A multifaceted approach was adopted to evaluate the performance of BBPP-SeNPs, encompassing antioxidant capacity, tyrosinase inhibition, and whitening effects, thereby furnishing robust experimental evidence for their potential applications.
However, there are areas that merit improvement:
1. While the introduction provides a comprehensive background, enhancing the logical flow by more explicitly connecting the biological activity of selenium to the specific context of this study could facilitate quicker comprehension of the research's motivation and significance.
2. For the particle size analysis (Figure 3.B), incorporating a graphical representation of particle size distribution over time or under varying synthesis conditions would more effectively illustrate the stability of BBPP-SeNPs.
3. The discussion section could benefit from a more pronounced emphasis on the unique advantages of BBPP-SeNPs relative to other similar materials, as well as a deeper exploration of the challenges and future research directions in practical applications. This would add greater depth and foresight to the article.
4. A more detailed safety evaluation of BBPP-SeNPs is warranted. While the high safety profile of nano selenium is mentioned, providing specific data on cell toxicity tests and long-term in vivo toxicity studies would strengthen the article's comprehensiveness.
5. Clarification is needed regarding Figure 3.C. Are the left and right images intended for comparison, and if so, do they represent a progressive relationship? Enhanced figure annotations would address these ambiguities.
Comments on the Quality of English LanguageEnglish writing is passable and can be understood by readers.
Round 2
Reviewer 2 Report
Comments and Suggestions for Authors
Dear Authors,
I appreciate the time and effort you have invested in addressing the corrections and questions raised during the review of your interesting work. Your responses are appropriate, and I am pleased to recommend your manuscript for publication in Molecules. Congratulations!
I would just suggest reviewing the format of the in-text citations, as they do not yet follow the standard format. I recommend checking the Molecules template, specifically the section on in-text citations. Typically, only the surname of the first author is mentioned, followed by et al. (without adding an apostrophe or 's' at the end), and then the reference number. For example: Wang et al. [1]. If there are up to three authors, all surnames should be included.
Best regards
